# Androgens Upregulate Pathogen-Induced Placental Innate Immune Response

**DOI:** 10.3390/ijms23094978

**Published:** 2022-04-29

**Authors:** Seline Vancolen, Taghreed Ayash, Mariela Segura, Marie-Julie Allard, Bernard Robaire, Guillaume Sébire

**Affiliations:** 1Department of Pharmacology & Therapeutics, McGill University, Montreal, QC H4A 3J1, Canada; seline.vancolen@mail.mcgill.ca (S.V.); bernard.robaire@mcgill.ca (B.R.); 2Department of Pediatrics, Research Institute of the McGill University Health Center, Montreal, QC H4A 3J1, Canada; taghreed.ayash@mail.mcgill.ca (T.A.); marie-julie.allard@mail.mcgill.ca (M.-J.A.); 3Faculty of Veterinary Medicine, University of Montreal, Montreal, QC H4A 3J1, Canada; mariela.segura@umontreal.ca

**Keywords:** androgens, autism spectrum disorder, cerebral palsy, chorioamnionitis, cytokines, group B *Streptococcus*, maternal immune activation, neurodevelopmental disorders, neonatal infection, placenta

## Abstract

Group B *Streptococcus* (GBS) is a leading cause of placental infection, termed chorioamnionitis. Chorioamnionitis is associated with an increased risk of neurobehavioral impairments, such as autism spectrum disorders, which are more prominent in males than in female offspring. In a pre-clinical model of chorioamnionitis, a greater inflammatory response was observed in placenta associated with male rather than female fetuses, correlating with the severity of subsequent neurobehavioral impairments. The reason for this sex difference is not understood. Our hypothesis is that androgens upregulate the placental innate immune response in male fetuses. Lewis dams were injected daily from gestational day (G) 18 to 21 with corn oil (vehicle) or an androgen receptor antagonist (flutamide). On G 19, dams were injected with saline (control) or GBS. Maternal, fetal sera and placentas were collected for protein assays and in situ analyses. Our results showed that while flutamide alone had no effect, a decrease in placental concentration of pro-inflammatory cytokines and infiltration of polymorphonuclear cells was observed in flutamide/infected compared to vehicle/infected groups. These results show that androgens upregulate the placental innate immune response and thus may contribute to the skewed sex ratio towards males observed in several developmental impairments resulting from perinatal infection/inflammation.

## 1. Introduction

The immune system plays a key role in the establishment, maintenance, and completion of a healthy pregnancy. Among other factors, it relies on a balance between pro- and anti-inflammatory cytokines and chemokines [1]. The dysregulation of genes involved in the inflammatory response, referred to as maternal immune activation (MIA), can cause an imbalance in this ratio. MIA often resulting from bacterial drivers, such as group B *Streptococcus* (GBS), can affect the highly orchestrated and vulnerable process of fetal brain development [2]. GBS is an opportunistic bacterium that colonizes the lower genital tract in 15–40% of pregnant women and is one of the leading causes of placental inflammation, termed chorioamnionitis [3,4]. A common intra-uterine infection scenario is the ascending route, where bacteria move chronologically from the vagina, through the cervix to penetrate the membrane and amniotic fluid [5]. GBS takes advantage of its adherence to luminal epithelial cells and surface proteins to invade the uterine cavity. Whatever the route of infection, which can be hematogenous contamination, chorioamnionitis is responsible for approximately 40% of preterm labor cases [6]. In addition to preterm birth, chorioamnionitis is also a risk factor for neurodevelopmental disorders, such as cerebral palsy (CP) and autism spectrum disorders (ASD) [4,7]. Importantly, CP and ASD are more frequent in males, with a skewed sex ratio of 4:1 [8,9]. While some genetic risk factors contribute to such male bias in perinatal neuro-disabilities, the cause for such a sex difference needs to be clarified [10].

Our laboratory has developed a GBS-induced chorioamnionitis rat model. In this model, GBS infection induced histological chorioamnionitis, as characterized by the presence of GBS and PMN, infiltrates within the decidua, labyrinth and amnion compartments of the placenta [4]. Particularly, PMN infiltration throughout placental tissue is the cardinal feature for the diagnosis of histological chorioamnionitis. In this model, a sex dichotomic inflammatory response was displayed, which was more intense in placenta associated with male rather than female fetuses and was correlated with the severity of subsequent neurobehavioral outcomes [11]. The reason for the difference in immune response between males and females is unclear. This knowledge gap raises the question of the role of sex steroids in the immune response. Testosterone, a steroid hormone produced by the gonads, possesses important developmental effects in utero. Testosterone interacts with the androgen receptor (AR) that is expressed in immune cells involved in chorioamnionitis, including polymorphonuclear cells (PMN) [12]. Between gestational day (G) 15 to G19, male fetal rats produce a peak of masculinizing testosterone, coinciding with the period of highest incidence of GBS-induced chorioamnionitis [13]. Together, these observations have led us to hypothesize that androgens play a key role in upregulating innate immunity in the context of bacterial infections of the placenta. In this study, our aim is to investigate if the androgen blockade results in a weaker inflammatory response invoked in males. We also investigate the effect of the androgen receptor antagonist, flutamide, in alleviating the molecular (pro-inflammatory cytokines) and cellular (PMN) responses in GBS-induced placental injury of males.

## 2. Results

### 2.1. Flutamide Acted as an Antiandrogen by Reducing AGD in Pups

We found that the dose of flutamide used counteracted the androgen effect on AGD in males (Figure 1a). AGD, defined as the distance between the anus and the genitals, is a sensitive biomarker for fetal androgen action. Alterations in this androgen-dependent developmental marker indicated that flutamide effectively induced androgen blockade. There was no effect of flutamide on the number of live fetuses per litter (Figure 1b).

### 2.2. Mild GBS Infection of Dams Induced in Our Model Was Well Tolerated Based on the Monitoring of Maternal Weight and Pro-Inflammatory Response in the Blood Circulation

GBS-infected dams displayed a stable weight gain compared to control except for a slight weight loss in the GBS plus flutamide group (*p* = 0.03) (Figure 2a). In the same line, there was no significant change of blood titers of IL-1β (*p* = 0.16) and IL-6 (*p* = 0.08) between experimental groups (Figure 2b,c).

### 2.3. Flutamide Decreased the Levels of Placental Proinflammatory Cytokines IL-1β, IL-6 and TNF-α in GBS-Induced Chorioamnionitis

We measured placental IL-1β, given that it is a key mediator of the pro-inflammatory response in GBS-induced chorioamnionitis [4]. GBS-infected placentas displayed increased levels of IL-1β compared to controls (Figure 3a). In the flutamide/GBS-exposed group, a significant decrease of IL-1β titers was detected compared to GBS-exposed placentas (Figure 3a). We also investigated other pro-inflammatory cytokines involved in GBS-induced chorioamnionitis, including IL-6 and TNF-α. Similarly, the significant increases of IL-6 and TNF-α concentrations in GBS-infected placenta were reversed in the GBS/flutamide group (Figure 3b,c). Flutamide on its own had no effects on any pro-inflammatory cytokine we examined.

### 2.4. Flutamide Reduced the Infiltration of PMN Cells in GBS-Infected Placentas

We quantified PMN densities in three placental regions: the decidua, junctional zone and labyrinth (Figure 4a). Massive PMN infiltrates were detected in the decidua of GBS-infected placentas compared to controls (Figure 4b and Figure 5a). GBS/flutamide placentas displayed a reduction in PMN infiltrates in the decidua (2.3 fold decrease) and labyrinth (1.7 fold decrease) compared to GBS-infected placentas (Figure 4a and Figure 5a,c). A trend (*p* = 0.07) toward such reduction in PMN infiltrates was detected in the junctional zone (Figure 4 and Figure 5b), while flutamide alone showed no effect.

## 3. Discussion

In line with our hypothesis, our results show that the androgen blockade significantly downregulated key inflammatory pathways involved in the placental response to GBS infection in male offspring, namely IL-1β release, downstream pro-inflammatory cytokines release such as TNF-α and IL-6, as well as PMN infiltration in both maternal and fetal compartments of the placenta.

This study shows that androgens have a pro-inflammatory effect in the placenta. It has been documented that sex hormones, such as androgens, mediate some sex-differences in the immune response [14,15]. However, there is conflicting evidence in the literature for the effect of testosterone on the immune system, as testosterone has been associated with both pro- and anti-inflammatory actions. For example, wound healing in males is impaired due to the pro-inflammatory action of testosterone upregulating cytokines such as TNF-α [16]. Another study in support of the pro-inflammatory actions of androgen used an AR knockout mice model. It was found that PMN cells were underrepresented among blood leukocytes from AR knock-out mice, suggesting that the androgen pathway could play an important role in maintaining an adequate circulating PMN level in vivo [17]. The opposite response is observed in testosterone-treated adult animals after traumatic injuries, i.e., depressed immune responses were noted [18]. Thus, it appears that the specific inflammatory effects induced by testosterone depends either on the animal model or stage of development [16].

The pro-inflammatory effect of androgen on the placenta might account for the skewed sex ratio towards males observed in perinatal neurobehavioral conditions such as CP and ASD. It may have to be considered in the design of sex specific placento- and neuro-protective anti-inflammatory treatment administered perinatally. ARs are located in a wide range of tissues including the testes, epididymes, prostate, seminal vesicles, muscle, brain, and from G13 to P10 in urogenital sinus and Wolffian ducts from G13 to P10 [19]. The AR is also detected at low levels in the rat placenta; however, the role of androgen signaling in the placenta has yet to be elucidated [20]. Goto et al. found that male rat offspring exposed to flutamide in utero displayed impairments of sexual behavior, cryptorchidism and the absence of the prostate gland and seminal vesicles in doses >10 mg/kg [21]. AGD at birth is a sensitive sex-related endpoint. In this study, the AGD of male pups from flutamide-treated mothers was markedly reduced, thus providing evidence for the action of flutamide on ARs in the male reproductive tract. It cannot be ascertained whether flutamide is acting on AR in the placenta, and if so on which resident and/or invading placental cells, but it is an interesting avenue for further research. In our model of live GBS inoculation at G19, the binding of testosterone on AR may activate both PMN proliferation and activation, including the upregulation of pro-inflammatory cytokines, such as IL-1. This, in turn, may overpass its protective anti-bacterial effects and induce placental and fetal collateral injuries. Hence, the interaction between testosterone and its cognate receptor in PMN cells might exacerbate the GBS-induced placental innate immune response. However, the mechanism by which androgen blockade downregulates the innate inflammatory reactions during chorioamnionitis is unclear. NF-KB is a transcription factor which plays a pivotal role in mediating the inflammatory response and is involved in IL-1 synthesis [22]. It would be interesting to investigate the effect of androgen on this main pathway involved in IL-1 synthesis using human neutrophil and macrophage cell lines. An additional interesting research avenue would be to investigate patterns of brain injuries and subsequent neurobehavioral abnormalities across treatment groups in this model of GBS-induced chorioamnionitis.

## 4. Materials and Methods

### 4.1. Bacteria and Flutamide Preparation

A stock tube of GBS serotype Ia (strain #16955) was stored in −80 °C in Brain Heart Infusion (BHI) broth with 15% glycerol and used for all experiments as previously described [4]. Briefly, GBS was inoculated into sterile BHI broth and incubated at 37 °C with shaking at 240 rotations per minute (rpm) for 18 h. The following day, GBS was reinoculated into sterile BHI broth and incubated under the same conditions. Optical density (OD) of the GBS plus BHI solution was measured frequently until the absorbance reached between 0.55–0.7 (OD_600 nm_). Once the desired absorbance was reached, 20 mL of the solution was centrifuged, and precipitated bacteria were washed twice in 20 mL of sterile 0.9% saline. The result was 1 mL of GBS plus saline per two microtubes, of which one was used for infections and the other for serial dilutions to determine the exact injected dose. GBS plus saline aliquots were kept on ice prior to injections. Serial dilutions from 10^−6^ to 10^−9^ were plated in triplicate on BHI agar plates and incubated overnight at 37 °C. The next morning, bacterial counts were determined by counting colonies to determine the exact injected dose: ~10^8^ CFU per 100 µL. Positive and negative controls were done on BHI agar plates and CHROMID Strepto B plates (BioMérieux, Saint-Laurent, QC, Canada), a screening media on which GBS colonies appeared red.

Flutamide is a competitive androgen receptor antagonist that acts by inhibiting the uptake and binding of androgens—both testosterone and its active 5α-reduced metabolite, dihydrotestosterone—to its receptor. Flutamide (Sigma-Aldrich, Oakville, ON, Canada) was dissolved in dimethyl sulfoxide (DMSO). The concentration of DMSO was adjusted to 5% by adding corn oil. Flutamide was administered at a dose of 50 mg/kg bodyweight and prepared on the same day as the injection. This dose was selected as it is well-established in impairing male reproductive development and preventing masculinization of anogenital distance (AGD) [23,24,25]. The vehicle control group was injected with 5% DMSO/corn oil.

### 4.2. End Gestational GBS Model and Injection Protocol

Primiparous Lewis dams were obtained from Charles River Laboratories (Kingston, NY) and arrived at the animal facility (RI-MUHC Glen Site, Montreal, QC, Canada) on G13. They were individually housed in a controlled environment (12 h light/dark cycle) and provided with water and food *ad libitum*. Dams were injected intraperitoneally every 24 h from G18 to G21 with 150 μL of either vehicle control or flutamide (Figure 1). On G19, dams were injected with 100 μL of sterile 0.9% saline or GBS serotype Ia suspended in saline (Figure 1), according to the experimental design we established in our laboratory [4,11]. All injections were done with one-hour intervals to avoid a time effect between inoculated dams. The four experimental groups (*n* litters per group) were: (1) vehicle plus control (*n* = 6), (2) vehicle plus GBS (*n* = 5), (3) flutamide plus control (*n* = 5) and flutamide plus GBS (*n* = 5) (Figure 6). All procedures were done in accordance with the guidelines established by the Canadian Council on Animal Care and were approved by the Research Institute of McGill University Health Centre (RI-MUHC, 2015-7675).

### 4.3. Cesarean (C)-Sections

Dams were weighed daily to determine the appropriate volume for flutamide injections and were observed to detect any abnormal behaviors. C-sections were done under general anesthesia (isoflurane 2%), according to a protocol routinely used in our laboratory [4], the day before natural birth (G22) with respect to their injection time (72 h post-GBS inoculation). Only placentas from live fetuses were included in the study.

### 4.4. Tissue Collection and Processing

While dams were under anesthesia, placentas and fetuses were removed. The position of each fetus in the uterine horn was noted. Differences in response to GBS infection according to fetal positioning were not observed [4]. Anogenital distance (AGD) was measured using digital calipers to determine fetal sex in non-flutamide groups and to confirm that the dose of flutamide used was effective by reducing the AGD in males. For flutamide groups, dissections were done to determine the sex of the fetus. Placentas were cut in half at the median coronal section, one half was fast-frozen (dipped into a jar of 2-methylbutane and stored on dry ice) and kept at −80 °C, while the other half was fixed in 4% buffered formaldehyde (0.1% glutaraldehyde, pH 7.4), processed and paraffin-embedded. In addition to placenta collection, maternal blood was also collected by heart puncture in Lithium Heparin Gel Separator tubes (BD Microtainer blood collection tubes, NJ). These tubes were centrifuged (13 min, speed 13,000 rpm), aliquoted and stored at −80 °C.

### 4.5. Placental Protein Extraction and Quantification

Proteins were extracted from placentas using the BCA Protein Assay Kit (Thermo Scientific-Pierce, Toronto, ON, Canada) and quantified using the Bradford Protein Assay (Bio-Rad, Mississauga, ON, Canada). Protein suspensions were then aliquoted and stored at −80 °C. Pro-inflammatory cytokines (IL-1β, IL-6, and TNF-α (R&D System, Minneapolis, MN, USA) from placenta protein suspensions and sera samples were measured using ELISA following the manufacturer’s protocol. All samples were measured in duplicates and the mean of the duplicates was used in analyses.

### 4.6. Immunohistochemistry and Image Analysis

In situ analyses were done as previously described [4]. Briefly, GBS serotype Ia and PMNs staining were investigated by immunohistochemistry (IHC). A rabbit GBS serotype Ia group-B typing antisera set (270,023 Ia, 1:500; Denka Seiken Co., Tokyo, Japan) was used to determine GBS infiltrates and a rabbit anti-PMN antibody (CLA51140, 1:100; Cedarlane Lab, Burlington, ON, Canada) was used to identify PMN infiltrates in maternal (decidua), junctional, and fetal (labyrinth) placental compartments. The secondary antibody used was a mouse horseradish peroxidase (HRP)-conjugated anti-rabbit (sc-2357, 1:100, Santa Cruz Biotechnology, Dallas, TX, USA).

A NanoZoomer Digital Pathology (NDP) Scanner (NanoZoomer 2.0-RS, Hamamatsu Photonics) was used to scan slides. The densities of PMNs in three placental compartments were determined as described previously [4]. Briefly, PMNs were counted in five pre-determined fields in the junctional and labyrinth areas, and three fields in the decidual area. This was done by an investigator blinded to the experimental groups. The density was calculated by dividing the number of counted cells by the field area. The mean density was used in the analyses. Only placentas developing chorioamnionitis—defined by the presence of GBS infiltrates in the decidua, junctional zone and labyrinth determined through IHC—were included in the study.

### 4.7. Statistical Analysis

One male per dam was used as *n* = 1 per litter to avoid artificial sample size inflation. Statistical analyses and figure representation were done using Graph Pad Prism software version 9.1.2 (San Diego, CA, USA). The assessment of normality and homogeneity of variance for each data set was conducted by Shapiro-Wilk normality test. When data were normally distributed, one-way ANOVA was applied with Tukey’s or Dunnett’s multiple comparisons test. When data were not normally distributed, a Kruskal-Wallis test was used, followed by Dunn’s multiple comparison test. Data are presented as the mean ± SEM; *p* ≤ 0.05 was considered statistically significant.

## 5. Conclusions

These results generate insights into the influence of sex steroids on innate immune responses in pathogen-induced placental injuries. Specifically, results show that androgens modulate the placental innate immune response by upregulating proinflammatory cytokines synthesis and PMN infiltration induced by bacterial infection. This process may contribute to the skewed sex ratio towards males observed in many developmental impairments of perinatal origin. The intent of this study is not to propose the use of the androgen blockade as a therapeutic agent. Rather, these results aim to establish a proof of concept regarding the mechanistic interplay between androgens and inflammatory mediators possibly responsible for sex-dichotomic neurobehavioral outcomes. In the future, this may pave the way to the development of targeted anti-inflammatory treatments combined with antibiotics adapted to fetal sex to treat gestational bacterial infection.

## Figures and Tables

**Figure 1 ijms-23-04978-f001:**
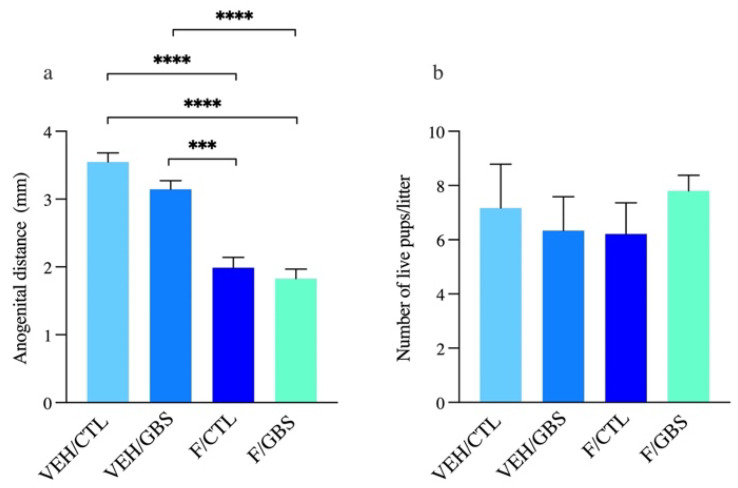
Flutamide acts as an antiandrogen by reducing AGD in pups. (**a**) The AGD of pups exposed in utero (G18–21) to flutamide or vehicle. (**b**) Number of live pups/litter. Analyses were done by one-way ANOVA with Tukey’s multiple comparison. Bars represent mean ± SEM. Number (*n*) of pups in vehicle/control: *n* = 6, vehicle/GBS: *n* = 5, flutamide/control: *n* = 5, flutamide/GBS: *n* = 5. One male was used per litter for analysis. *** *p* < 0.001, **** *p* < 0.0001.

**Figure 2 ijms-23-04978-f002:**
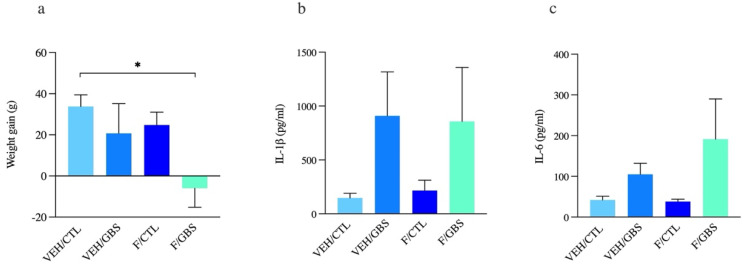
Monitoring of maternal weight and pro-inflammatory response in the blood circulation. Mean maternal weight gain from G18–22 (**a**). Mean concentrations of IL-1β (**b**) and IL-6 (**c**) in maternal sera. Analyses were done by one-way ANOVA with Tukey’s multiple comparison (maternal weight gain, IL-1β) or Kruskal-Wallis with Dunn’s multiple comparison test when data was not normally distributed (IL-6). Bars represent mean ± SEM. Number (*n*) of dams in vehicle/control: *n* = 6, vehicle/GBS: *n* = 5, flutamide/control: *n* = 5, flutamide/GBS: *n* = 5. * *p* < 0.05.

**Figure 3 ijms-23-04978-f003:**
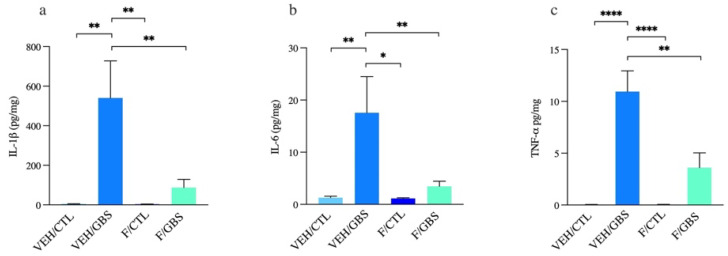
In combination with flutamide, GBS-induced chorioamnionitis is associated with decreased levels of placental proinflammatory cytokines IL-1β, IL-6 and TNF-α. Mean placental concentrations of (**a**) IL-1β, (**b**) IL-6, (**c**) TNF-α detected by ELISA post-GBS inoculation. Analyses were done by one-way ANOVA with Dunnett’s multiple comparison (to vehicle/GBS). Bars represent mean ± SEM. Number (*n*) of placentas in vehicle/control: *n* = 6, vehicle/GBS: *n* = 5, flutamide/control: *n* = 5, flutamide/GBS: *n* = 5. One male placenta was used per litter. * *p* < 0.05, ** *p* < 0.01, **** *p* < 0.0001.

**Figure 4 ijms-23-04978-f004:**
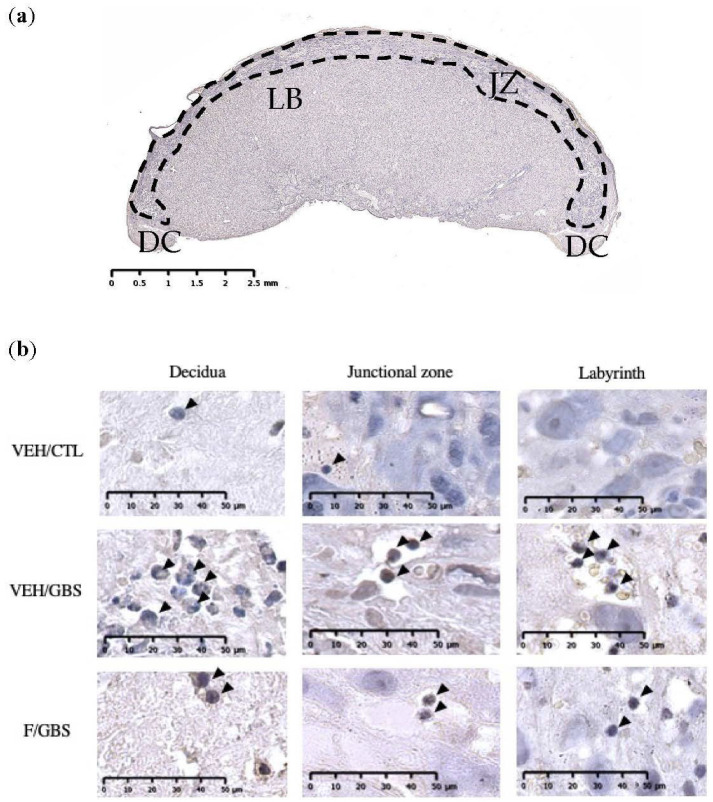
Flutamide reduced the infiltration of PMN cells in GBS infected placental compartments. (**a**) Coronal section of placental compartments showing the maternal (decidua, DC), maternofetal zone (junctional zone, JZ), and fetal (labyrinth, LB) compartments. (**b**) PMN infiltrations (circular in shape, stained dark purple/brown staining, black arrowheads) in DC, JZ and LB compartments of the placenta.

**Figure 5 ijms-23-04978-f005:**
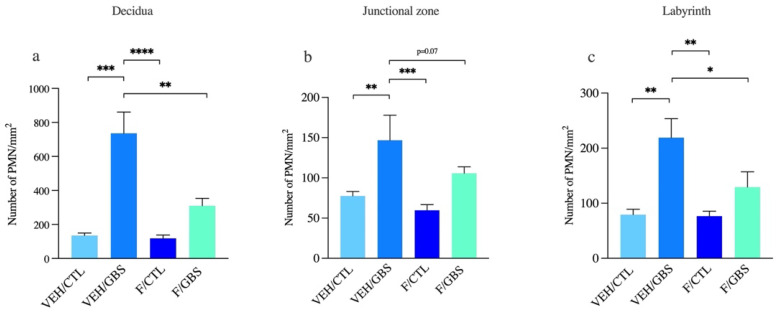
Mean density of PMNs detected in GBS-infected placental compartments. (**a**) decidua (DC) (**b**) junctional zone (JZ), and (**c**) fetal labyrinth (LB). Statistical analyses were done by one-way ANOVA with Dunnett’s multiple comparison (to vehicle/GBS). Bars represent mean ± SEM. Number (*n*) of pups in vehicle/control: *n* = 4–5, vehicle/GBS: *n* = 3–6, flutamide/control: *n* = 5–6, flutamide/GBS: *n* = 6. One male was used per litter for analysis. * *p* < 0.05, ** *p* < 0.01, *** *p* < 0.001, **** *p* < 0.0001.

**Figure 6 ijms-23-04978-f006:**
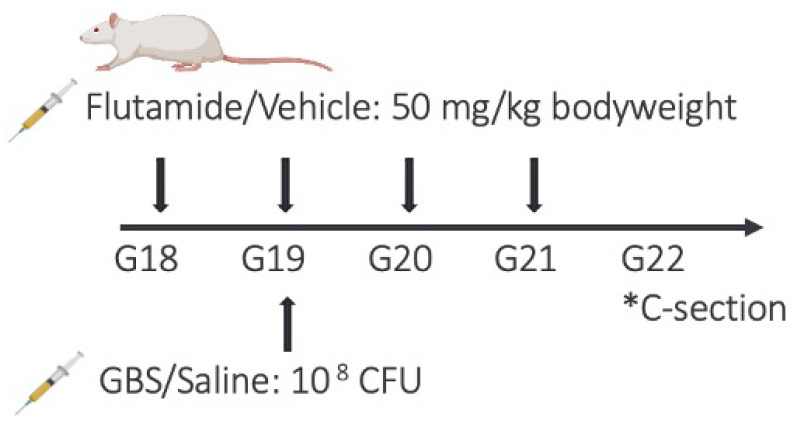
Timeline of injections and C-section of dams from G18 to G21.

## Data Availability

Not applicable.

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
