# Peer review of "Androgens Upregulate Pathogen-Induced Placental Innate Immune Response"

_ijms, 2022, doi:10.3390/ijms23094978_

Round 1

Reviewer 1 Report

Within this manuscript, the authors have documented androgens role in upregulating immune cell infiltration and inflammation of the placenta in a rat model. There are, however, some key points that need to be addressed before publication.

  • Fundamentally, this GSB model of immune cell infiltration to the placenta is improperly classified. This is not a model of “chorioamnionitis,” as it says throughout the abstract, intro, results, and discussion of this manuscript. Chorioamnionitis is clinically defined as immune cell infiltration at the decidua parietalis-chorion interface of the fetal membrane, not the decidua basalis-placenta. The model the authors have developed mimics “placentitis,” which is clinically very rare and does not often form during GBS infection. Therefore, the authors should change the description of their model throughout the text of the manuscript in order to properly describe the pathology they have induced with GBS in this rat model.
  • In figure 4, the placenta cell morphology of the decidua, junctional zone, and labyrinth should be included to confirm the image location.
  • Since the authors collected the fetuses from these experiments, it would be nice to see if the male fetuses had any adverse neuronal outcomes compared to females.

Reviewer 2 Report

This study investigates the role of androgens in mediationg pathogen induced innate immune response in the placenta. The authors found that androgen blockade down regulates immune processes involved in placental response to GBS infection. It is a well written manuscript with clearly presented methodology. I have the following comments:

  1. In the introduction section the authors correctly indicate that GBS colonises lower genital tract of 15-40% of pregnant women. At this point the authors should further discuss the reasons for ascending infection, and the percentage of preterm births caused by chorioamnionitis.
  2. How whould the authors set up a clinical study to determine the possible role of androgens in placental immune response to various pathogens?
